

# Prognostic analysis of very early onset pancreatic cancer: a population-based analysis

Dongjun Dai[1], Yanmei Wang[1], Xinyang Hu[1], Hongchuan Jin[2] and Xian Wang[1]

[1] Department of Medical Oncology, Sir Run Run Shaw Hospital, Medical School of Zhejiang University, Zhejiang University, Hangzhou, Zhejiang, China
[2] Laboratory of Cancer Biology, Key Lab of Biotherapy, Sir Run Run Shaw Hospital, Medical School of Zhejiang University, Zhejiang University, Hangzhou, Zhejiang, China

## ABSTRACT

**Background**. We aimed to use competing risk model to assess whether very early onset pancreatic cancer (VEOPC ) (<45 years) had a worse prognosis than older pancreatic cancer (PC) patients, and to build a competing risk nomogram for predicting the risk of death of VEOPC.

**Methods**. We selected pancreatic adenocarcinoma (PDAC) patients as our cohort from the Surveillance, Epidemiology, and End Results (SEER) database. The impact of cancer specific death was estimated by competing risk analysis. Multivariate Fine-Gray regression for proportional hazards modeling of the subdistribution hazard (SH) model based nomogram was constructed, which was internally validated by discrimination and calibration with 1,000 bootstraps.

**Results**. Our cohort included 1,386 VEOPC patients and 53,940 older patients. We observed that in unresectablePDAC patients, VEOPC had better cancer specific survival (CSS) than each older group (45–59 years, 60–69 years, 70–79 years and >79 years). There was no significant prognostic difference between VEOPC and each older group in resectablePDAC. Our competing nomogram showed well discrimination and calibration by internal validation.

**Conclusion**. For unresectable PDAC patients, VEOPC had better CSS than older patients. Our competing risk nomogram might be an easy-to-use tool for the specific death prediction of VEOPC patients with PDAC.

## INTRODUCTION

Pancreatic cancer (PC) patients have a 5-year survival rate of lower than 10% (*Siegel, Miller & Jemal, 2018*). According to the American Cancer Society, the average age of PC patients at diagnosis is 70 years. Less than 3% of PC patients are under the age of 45 years at diagnosis (*Surveillance, Epidemiology End Results Program, 2019a*), which was defined as very early onset PC (VEOPC) (*McWilliams et al., 2016*). Duo to its rareness, few data were reported regarding the clinicopathological features and treatment response of VEOPC

Corresponding author
Xian Wang, wangx118@zju.edu.cn

patients (*Grosfeld et al., 1990*; *Jaksic et al., 1992*; *Lack et al., 1983*; *McWilliams et al., 2016*; *Shorter et al., 2002*).

The identification of prognostic factors of PC might improve the prediction of the survival and the selection of therapy for PC patients. The well-known PC prognostic factors comprised resection margins, lymph nodes status and lymph node ratio, perineural and blood vessel invasion, tumor localization and body mass index (BMI) and treatment such as chemotherapy (*Bilici, 2014*; *Lin et al., 2016*). However, whether age is a prognostic factor of PC is currently controversial. Some studies found "younger" PC patients (with different definition of age <45 years, age <50 years, age <60 years and age <70 years) had better overall survival (OS) than the older PC patients (*Baxter, Whitson & Tuttle, 2007*; *He et al., 2013*; *Saad et al., 2018*; *Sugiura et al., 2017*). While the other studies found younger age (with different definition of age <65 years and age <70 years *Barbas et al., 2012*; *Miyazaki et al., 2016*; *Van der Geest et al., 2016*) was not associated with the survival of PC patients. It should be noted that these studies had a limited number of VEOPC patients or did not treat the VEOPC as a group.

Nomograms are a valuable tool to use different clinical variables to determine a statistical prognostic model that generates a probability of clinical outcomes for a single patient (*Balachandran et al., 2015*). Nomograms have been applied in various types of cancers (*Fakhry et al., 2017*; *Huang et al., 2016*; *Liang et al., 2015*; *Rose et al., 2015*). Nowadays, nomograms are commonly used to estimate prognosis in oncology and medicine, which fit the trend of personalized medicine (*Balachandran et al., 2015*). It is worth constructing a nomogram for predicting the prognosis of VEOPC patients.

Pancreatic adenocarcinoma (PDAC) is the main type of PC, which has a worse prognosis than other histology types. The current study selected PDAC patients as our cohort. It should be noted that the competing event of death is widely existing in the prognostic analysis of cancer. The failure to account for such competing events, such as the Kaplan–Meier method and standard Cox proportional hazards regression analysis, might result in an overestimate of the cumulative incidences (*Satagopan et al., 2004*). The competing risk model considers both the disease-specific death and other causes of death, which was fit to the analysis in the presence of competing risk events (*Kim, 2007*). The competing risk analysis was often missed in previous studies when included with very aged PC patients. The aim of our study was to use competing risk model to estimate the prognostic value of age in PDAC patients and construct competing nomogram for cancer–specific death prediction of VEOPC patients by using the data from the Surveillance, Epidemiology, and End Results (SEER) database of the National Cancer Institute of the United States, which currently collects and publishes cancer incidence and survival data from population-based cancer registries covering approximately 34.6 percent of the United States population (*Surveillance Epidemiology End Results Program, 2019b*).

## METHODS

### Cohort selection

The cohort was selected by SEER*Stat 8.3.5 software with the following criteria: (1) it should be a primary PDAC patients (ICD-O-3 histology codes 8140 and 8500) registered

equal or after year of 2004; (2) it should be a patient who has a survival time over than 0 month with records of survival status; (3) it should have at least one year follow up time to capture enough events to ensure a meaningful analysis. We also excluded few patients with unclear information of race, surgery, number of regional lymph nodes examined, radiotherapy and marital status. Finally, we excluded 1 patient with cancer in the islets of langerhans. The detailed data selection process was shown in Fig. S1.

## Study variables and endpoints

We included 14 variables into our analysis, which were listed as the follows: age at diagnosis, sex (Females and males), race (Caucasian, African American, American Indian/Alaska Native and Asian or Pacific Islander), tumor location (Head of pancreas, body of pancreas, tail of pancreas, pancreatic duct, other specified parts of pancreas, overlapping lesion of pancreas and Pancreas, NOS (Not otherwise specified)), surgery experience (No surgery, local or partial pancreatectomy, local or partial pancreatectomy and duodenectomy, total pancreatectomy with or without gastrectomy or duodenectomy, pancreatectomy NOS or surgery NOS), tumor size, LNR (Lymph nodes ratio, which is defined as the ratio of the number of positive lymph nodes to the total number of lymph nodes examined), 6th American Joint Committee on Cancer (AJCC) tumor TNM stage, grade, radiotherapy and chemotherapy experience and marital status. The value of age at diagnosis, tumor size and LNR were classified into small categorical variables to fit the linear assumption. The widowed or single (Never married or having a domestic partner) or divorced or separated patients was defined as "Single". The median observed survival time was referred to median follow-up time.

## Statistical analyses

The difference of each variable between VEOPC and other PC patients was evaluated by Chi-Squared tests. Cumulative incidences of death (CID) was estimated for deaths caused by cancer or other reasons by using the cumulative incidences function (CIF) analysis. Multivariate SH model was used to assess the cancer specific survival (CSS). All the variables were included in the multivariate analysis. Hazard ratio (HR) and 95%confidence index (95%CI) were calculated.

Multivariate SH model-based nomogram was constructed by multivariate logistic regression model to predict the 6 months, 12 months, 18 months and 24 months cancer specific death of VEOPC patients. The internal validation of nomogram was performed by discrimination and calibration with 1,000 time bootstraps (Balachandran et al., 2015). The discrimination was estimated by the area under the curve of receiver operating characteristic curve (AUC, which was also referred as C-statistics). The AUC ranges from 0.5–1.0, with 0.5 indicates the outcomes is totally random and 1.0 indicates the perfect discrimination. The calibration was assessed by calibration curves, which shows how close the nomogram estimated risk was to the observed risk. All the statistical analyses were performed by "R" version 3.6.0. The CIF test and multivariate SH analysis were performed by "R" package "cmprsk". The competing risk nomogram was constructed by "R" package "mstate" and plotted by "R" package "regplot". The calibration curve was drawn by "R" package

"riskRegression". The AUC was calculated by "R" package "riskRegression" and plotted by "R" package "ggplot2". A two tailed $p$-value less than 0.05 was considered statistically significant.

# RESULTS

## Cohort selection

We included the patients diagnosed equal to or before 2015 to ensure an enough follow-up time. There were 1,386 VEOPC patients and 53,940 older PDAC patients ($\geq$45 years). The median survival time of VEOPC and older patients were 9 and 7 months, respectively. Significant differences were found between VEOPC and other PC patients. The VEOPC patients had more males, fewer Caucasians, more surgery rates, fewer LNR, higher TNM stage, higher grade, more experience of chemotherapy and radiotherapy and fewer of married status ($p < 0.001$, Table 1) than the older patients. We also provided a table with characteristics for all patients (Table S1). We divided the older patients into 4 groups with different age ranges (45–59 years, 60–69 years, 70–79 years and >79 years) for further analysis.

## Prognostic analysis of age in PDAC patients

The CIF plot showed VEOPC patients had a decreased risk of cancer death than older patients ($p < 0.001$, Fig. 1A). Further subgroup analyses by surgery showed that VEOPC patients had better prognosis than the older patients whether they performed surgery or not ($p < 0.001$, Figs. 1B–1C). Moreover, multivariate SH model with all the included variables showed that in unresectable PDAC patients, the VEOPC had better CSS than older patients. While in the resected PDAC patients, no significant difference was found between VEOPC and older patients (Table 2).

## Independent prognostic factors of CSS of VEOPC

As shown in Table 3, multivariate SH model was used to find independent prognostic factors for the CSS of VEOPC patients. We found the female, race of Asian or Pacific Islander, PC located in pancreatic duct, no surgery, high LNR, high M stage, high grade, lack of chemotherapy, and single status were risk factors of the VEOPC patients ($P < 0.05$).

## Nomogram development and validation

The nomogram was constructed by multivariate SH model. A weighted total score calculated from each variable was used to estimate the 6 months, 12 months, 18 months and 24 months cancer specific death of VEOPC patients (Fig. 2). The sex, race, location, surgery strategy, tumor size, LNR, TNM stage, grade, chemotherapy and marital status were found to be significantly important for the prognosis of VEOPC patients (Variables with asterisk in Fig. 2). The calibration plots showed well correlation between observed CSS and nomogram predicted CSS (Figs. 3A–3D). Time dependent AUC plot suggested well discrimination of our nomogram (AUC $\leq$ 0.736, Fig. 3E). Furthermore, we equally stratified the whole VEOPC into three groups based on their level of competing nomogram scores and performed a CIF analysis. The CIF plot (Fig. 4) showed VEOPC with higher scores of

**Table 1  The demographic and treatment data of VEOPC and older PC (≥45 years) patients.**

| Characteristics | VEOPC (N = 1,386) | | Older PC (N = 5,3932) | |
|---|---|---|---|---|
| | No. of patients | % | No. of patients | % |
| **Sex** | | | | |
| Male | 796 | 57.4% | 27337 | 50.7% |
| Female | 590 | 42.6% | 26595 | 49.3% |
| **Race** | | | | |
| Caucasian | 1026 | 74.0% | 43089 | 79.9% |
| African American | 207 | 14.9% | 6522 | 12.1% |
| American Indian/Alaska Native | 13 | 0.9% | 287 | 0.5% |
| Asian or Pacific Islander | 140 | 10.1% | 4034 | 7.5% |
| **Location** | | | | |
| Head of pancreas | 738 | 53.2% | 29410 | 54.5% |
| Body of pancreas | 154 | 11.1% | 6921 | 12.8% |
| Tail of pancreas | 182 | 13.1% | 6458 | 12.0% |
| Pancreatic duct | 12 | 0.9% | 319 | 0.6% |
| Other specified parts of pancreas | 18 | 1.3% | 805 | 1.5% |
| Overlapping lesion of pancreas | 113 | 8.2% | 4049 | 7.5% |
| Pancreas, NOS | 169 | 12.2% | 5970 | 11.1% |
| **Surgery** | | | | |
| No surgery | 1040 | 75.0% | 41989 | 77.9% |
| Local or partial pancreatectomy | 43 | 3.1% | 1517 | 2.8% |
| Local or partial pancreatectomy and duodenectomy | 242 | 17.5% | 8175 | 15.2% |
| Total pancreatectomy with or without gastrectomy or duodenectomy | 50 | 3.6% | 1978 | 3.7% |
| Pancreatectomy NOS or surgery NOS | 11 | 0.8% | 273 | 0.5% |
| **Tumor size (cm)** | | | | |
| ≤2 | 144 | 10.4% | 4883 | 9.1% |
| 2 to 4 | 524 | 37.8% | 23278 | 43.2% |
| 4 to 6 | 333 | 24.0% | 12388 | 23.0% |
| >6 | 104 | 7.5% | 3840 | 7.1% |
| Unknown | 281 | 20.3% | 9543 | 17.7% |
| **LNR** | | | | |
| 0 | 137 | 9.9% | 4542 | 8.4% |
| ≤0.2 | 126 | 9.1% | 3983 | 7.4% |
| 0.2–0.4 | 57 | 4.1% | 2180 | 4.0% |
| 0.4–1 | 55 | 4.0% | 2116 | 3.9% |
| No nodes were examined | 1011 | 72.9% | 41111 | 76.2% |
| **T stage** | | | | |
| T1-T2 | 275 | 19.8% | 11774 | 21.8% |
| T3-T4 | 865 | 62.4% | 33301 | 61.7% |
| Unknown | 246 | 17.7% | 8857 | 16.4% |

**Table 1** (*continued*)

| Characteristics | VEOPC (N = 1,386) | | Older PC (N = 5,3932) | |
|---|---|---|---|---|
| | No. of patients | % | No. of patients | % |
| **N stage** | | | | |
| N0 | 603 | 43.5% | 26845 | 49.8% |
| N1 | 584 | 42.1% | 19325 | 35.8% |
| Unknown | 199 | 14.4% | 7762 | 14.4% |
| **M stage** | | | | |
| M0 | 582 | 42.0% | 26570 | 49.3% |
| M1 | 786 | 56.7% | 25678 | 47.6% |
| Unknown | 18 | 1.3% | 1684 | 3.1% |
| **Grade** | | | | |
| I-II | 326 | 23.5% | 11939 | 22.1% |
| III-IV | 256 | 18.5% | 8755 | 16.2% |
| Unknown | 804 | 58.0% | 33238 | 61.6% |
| **Chemotherapy** | | | | |
| No | 312 | 22.5% | 20586 | 38.2% |
| Yes | 1074 | 77.5% | 33346 | 61.8% |
| **Radiotherapy** | | | | |
| No | 1021 | 73.7% | 42283 | 78.4% |
| Yes | 365 | 26.3% | 11649 | 21.6% |
| **Marital status** | | | | |
| Married | 768 | 55.4% | 32013 | 59.4% |
| Single | 618 | 44.6% | 21919 | 40.6% |

**Notes.**
All variables showed significant differences between VEOPC and older PC patients ($p < 0.001$).

our competing nomogram was at high risk of cancer related death. It indicated that our competing nomogram was useful for risk stratification of the prognosis of VEOPC patients. The detailed competing nomogram scores for all variables were listed in Table S2).

## DISCUSSION

No previous competing analysis was performed to analyze the prognostic difference between VEOPC and older PC patients. The current study used competing risk model to estimate the prognostic difference between VEOPC and older patients by using the data from SEER. We found that among the unresectable PDAC patients, the VEOPC had better CSS than older patients. And among resected PDAC patients, there was no significant prognostic difference between VEOPC and older patients. We also built the first competing nomogram for primary VEOPC patients with histology type of PDAC.

Age was observed as an independent prognostic factor for many cancers. Previous studies identified that younger patients had worse prognosis than older patients in colorectal cancer (*Van Eeghen et al., 2015*) and breast cancer (*Chen et al., 2016*) while younger patients had better prognosis than older patients in lung cancer (*Arnold et al., 2016*), liver cancer (*Zhang & Sun, 2015*) and prostate cancer (*Pettersson et al., 2018*). A study found that the <45 years PDAC patients had better survival than >70 years PDAC patients (*He et al., 2013*). Another

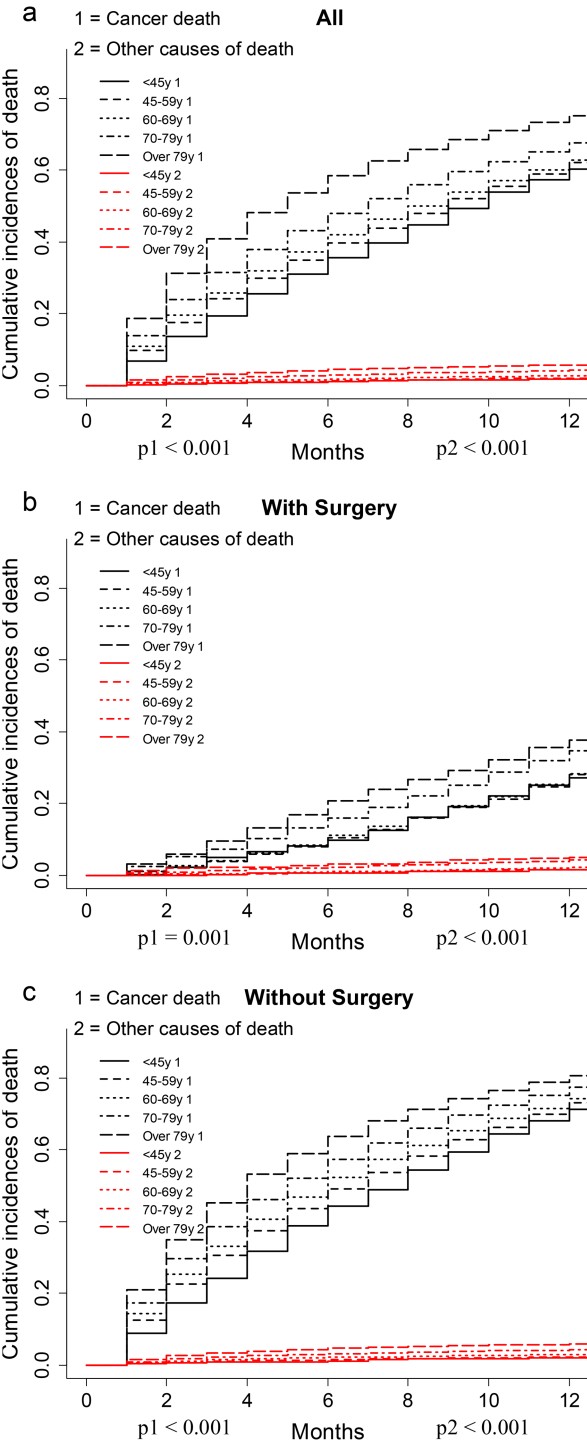

**Figure 1** **CIF plots of the association between age and the prognosis of PDAC patients.** (A–C) The CIF plots of the association between age and the prognosis of all cohort (A), patients with surgery (B) or without surgery (C).

**Table 2  Multivariate SH analysis of CSS for the association between age and the prognosis of PDAC patients.** We included all variables in the SH multivariate analysis.

| Age (Years) | For the whole cohort | | For patients with surgery | | For patients without surgery | |
|---|---|---|---|---|---|---|
| | HR (95% CI) | *p* value | HR (95% CI) | *p* value | HR (95% CI) | *p* value |
| <45 | Reference | | Reference | | Reference | |
| 45–59 | **1.08 (1.02–1.14)** | **0.010** | 0.98 (0.86–1.12) | 7.50E−01 | **1.10 (1.03–1.16)** | **0.003** |
| 60–69 | **1.10 (1.04–1.17)** | **<0.001** | 1.01 (0.89–1.15) | 9.00E−01 | **1.13 (1.06–1.20)** | **<0.001** |
| 70–79 | **1.16 (1.09[-1.23)** | **<0.001** | 1.04 (0.91–1.19) | 5.70E−01 | **1.19 (1.12–1.26)** | **<0.001** |
| >79 | **1.17 (1.10–1.24)** | **<0.001** | 1.00 (0.87–1.17) | 9.60E−01 | **1.19 (1.12–1.27)** | **<0.001** |

**Notes.**
HR, Hazard ratio; 95% CI, 95% confidence index.
Significant results ($p < 0.05$) were bolded.

study found <50 years PDAC patients had no significant difference on prognosis when compared with their whole cohort (*Tingstedt, Weitkamper & Andersson, 2011*). To be noted, these studies only included with limited samples. A SEER-based study found the risk of cancer-specific death decreases with age in resectable PDAC patients (*He et al., 2018*). However, we found the age was not an independent prognostic factor for resectable PDAC patients. This difference might be caused by the difference of inclusion criteria that our study included more variables in the multivariate SH model. We provided a detailed inclusion workflow in the Fig. S1. We found the VEOPC patients, despite with a higher TNM stage, had a better CSS than the older patients in unresectable patients (Table S1). It might be explained by the well accepted fact that younger patients had more tolerance to chemotherapy. Indeed, in the unresectable group, the VEOPC patients had a 17.5% higher rate to have chemotherapy than the older patients.

Males had higher rate of PC incidence in United States (*Yadav & Lowenfels, 2013*). We observed the males had 15% higher rate than the females in VEOPC. Previous study found males had worse prognosis than females in patients under 60 years but not in VEOPC patients (*McWilliams et al., 2016*). It might be caused by the sample size. Our study had much higher amount of VEOPC patients than this study and we found the males had worse CSS than females.

Marriage status was linked to the improvements in cardiovascular, endocrine, immune function, and cancer prognosis (*Aizer et al., 2013*; *Gallo et al., 2003*; *Herberman & Ortaldo, 1981*). Previous study observed that unmarried patients were at significantly higher risk of presentation with metastatic cancer, undertreatment, and death resulting from their cancer (*Aizer et al., 2013*). We found the VEOPC patients with married status had better CSS than patients with single status, which provided new hint to the association between support care and cancer prognosis.

Our SH model based nomogram was well validated by discrimination and calibration. The variables included in our nomogram were easy to be obtained, allowing a feasible translation into the future clinical use. To be noted, we included both the significant variables and the insignificant variables into our nomogram, since if a nomogram only includes statistically significant variables, it might tend to exert an inappropriately large influence, leading to falsely narrowed confidence intervals, which would make this

**Table 3  Multivariate SH analyses of CSS for each variable in VEOPC patients.** We included all variables in the SH multivariate analysis.

| Characteristics | HR (95% CI) | p value |
|---|---|---|
| **Sex** | | |
| Male | Reference | |
| Female | **0.87 (0.77–0.98)** | **0.021** |
| **Race** | | |
| Caucasian | Reference | |
| African American | 1.02 (0.84–1.23) | 0.880 |
| American Indian/Alaska Native | 1.00 (0.67–1.50) | 1.000 |
| Asian or Pacific Islander | **1.54 (1.30–1.83)** | **<0.001** |
| **Location** | | |
| Head of pancreas | Reference | |
| Body of pancreas | 0.89 (0.73–1.08) | 0.220 |
| Tail of pancreas | 1.07 (0.87–1.32) | 0.530 |
| Pancreatic duct | **2.01 (1.34–3.02)** | **0.001** |
| Other specified parts of pancreas | 0.65 (0.33–1.29) | 0.220 |
| Overlapping lesion of pancreas | 0.92 (0.72–1.18) | 0.520 |
| Pancreas, NOS | **1.28 (1.03–1.60)** | **0.025** |
| **Surgery** | | |
| No surgery | Reference | |
| Local or partial pancreatectomy | **0.40 (0.26–0.61)** | **<0.001** |
| Local or partial pancreatectomy and duodenectomy | **0.46 (0.36–0.60)** | **<0.001** |
| Total pancreatectomy with or without gastrectomy or duodenectomy | **0.41 (0.28–0.61)** | **<0.001** |
| Pancreatectomy NOS or surgery NOS | 0.77 (0.36–1.64) | 0.490 |
| **Tumor size** | | |
| ≤2 cm | Reference | |
| 2 to 4 | 1.07 (0.87–1.32) | 0.510 |
| 4 to 6 | 1.11 (0.88–1.41) | 0.360 |
| >6 | 1.28 (0.94–1.74) | 0.110 |
| Unknown | 1.01 (0.78–1.32) | 0.930 |
| **LNR** | | |
| 0 | Reference | |
| ≤0.2 | **1.39 (1.02–1.90)** | **0.037** |
| 0.2–0.4 | **1.80 (1.30–2.51)** | **<0.001** |
| 0.4–1 | **1.90 (1.37–2.64)** | **<0.001** |
| No nodes were examined | **1.33 (1.04–1.70)** | **0.021** |
| **T stage** | | |
| T1-T2 | Reference | |
| T3-T4 | 1.16 (0.98–1.37) | 0.082 |
| Unknown | 0.95 (0.73–1.22) | 0.680 |

**Table 3** (*continued*)

| Characteristics | HR (95% CI) | *p* value |
|---|---|---|
| **N stage** | | |
| N0 | Reference | |
| N1 | 1.12 (0.95–1.31) | 0.170 |
| Unknown | **1.34 (1.09–1.66)** | **0.005** |
| **M stage** | | |
| M0 | Reference | |
| M1 | **1.48 (1.25–1.75)** | **<0.001** |
| Unknown | 1.09 (0.60–1.96) | 0.780 |
| **Grade** | | |
| I-II | Reference | |
| III-IV | **1.29 (1.07–1.56)** | **0.007** |
| Unknown | 1.09 (0.93–1.29) | 0.290 |
| **Chemotherapy** | | |
| No | Reference | |
| Yes | **0.63 (0.53–0.76)** | **<0.001** |
| **Radiotherapy** | | |
| No | Reference | |
| Yes | 1.09 (0.94–1.25) | 0.250 |
| **Marital status** | | |
| Married | Reference | |
| Single | **1.15 (1.02–1.30)** | **0.021** |

**Notes.**

HR, Hazard ratio; 95% CI, 95% confidence index.
Significant results ($p < 0.05$) were bolded.

nomogram appear more accurate than it is (*Vogelzang et al., 2005*). Patients with higher competing nomogram scores of our study were found to have higher CID, indicating that our nomogram was useful for risk stratification of the prognosis of VEOPC patients.

Our study had some limitations. First, the detailed strategy of chemotherapy or radiotherapy was missed in current study as well as other therapies such as immunotherapy. Second, although the current nomogram was well validated by internal validation, there was a lack of an external validation by cohort other than SEER. Third, it should be noted that only factors available in the SEER registry were examined. The results from this study might only fit the population from United States. Fourth, our study has no exploration of molecular mechanism, which was of interest but cannot be conducted at this time with data currently available.

# CONCLUSIONS

The current study showed that for unresectable PDAC, VEOPC had better CSS than older patients. For the VEOPC patients, we built a SH model based nomogram to estimate their CSS. This nomogram was shown to have good discrimination and calibration.

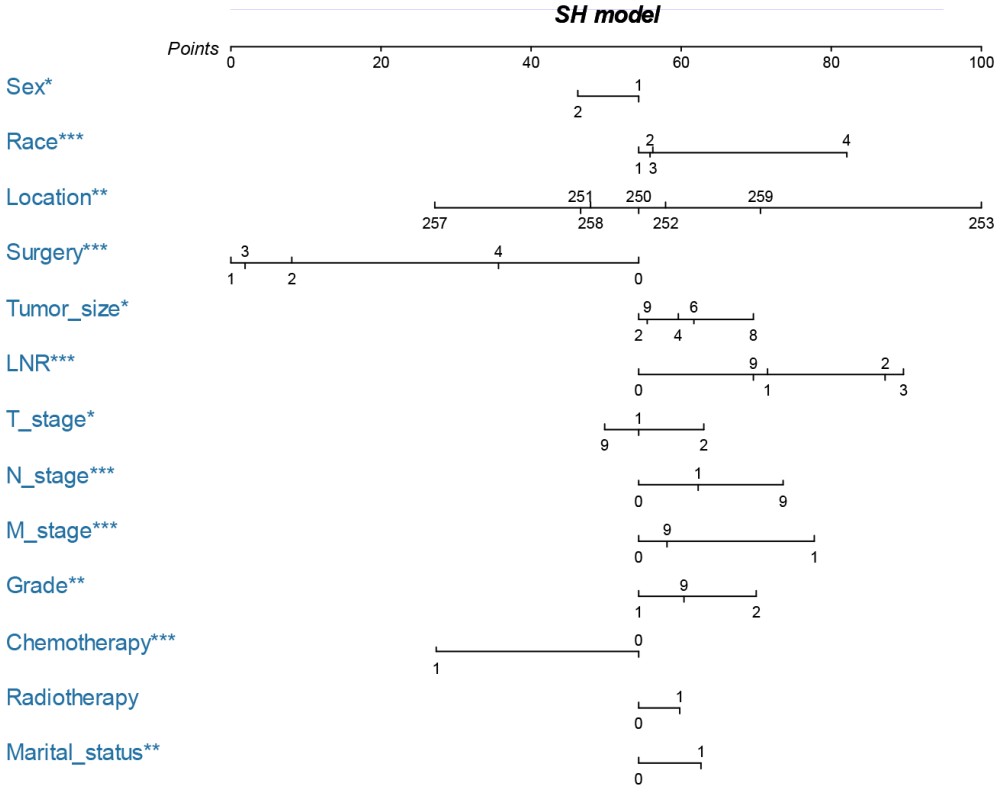

**Figure 2  Nomogram for predicting 6 months, 12 months, 18 months and 24 months cancer-specific death of VEOPC patients.** The nomogram is used by summing the points identified on the top scale for each independent variable and drawing a vertical line from the total points scale to the 6 months, 12 months, 18 months and 24 months cancer specific death to obtain the probability of survival. The total points projected to the bottom scale indicate the probability of the 6 months, 12 months, 18 months and 24 months cancer specific death. Sex: 1 = Male, 2 = Female; Race: 1 = Caucasian, 2 = African American, N = Other race; Location: 250 = Head of pancreas, 251 = Body of pancreas, 252 = Tail of pancreas, 253 = Pancreatic duct, 257 = Other specified parts of pancreas, 258 = Overlapping lesion of pancreas, 259 = Pancreas, NOS; Surgery, 0 = No surgery, 1 = Local or partial pancreatectomy, 2 = Local or partial pancreatectomy and duodenectomy, 3 = Total pancreatectomy with or without gastrectomy or duodenectomy, 4 = Pancreatectomy NOS or surgery NOS; Tumor size: 2 = <two cm, 4 = two cm to four cm, 6 = four cm to six cm, 8 = >6 cm, 9 = Unknown; LNR: 0 = 0, 1 = <0.2, 2 = 0.2–0.4, 3 = >0.4, 4 = No nodes were examined; T stage: 1 = T1-T2, 2 = T3-T4, 9 = Unknown; N stage: 0 = NO, 1 = N1, 9 = Unknown; M stage: 0 = M0, 1 = M1, 9 = Unknown; Grade: 1 = I-II, 2 = III-IV, 9 = Unknown; Chemotherapy: 0 = none/unknown and 1 = yes; Radiotherapy: 0 = none/unknown or refused, 1 = beam radiation or combination of beam with implants or isotopes or radiation with method or source not specified or radioactive implants or radioisotopes and N = Recommended, unknown if administered; Marital status: 0 = married; 1 = widowed or single (never married or having a domestic partner) or divorced or separated. The asterisk indicates significance.

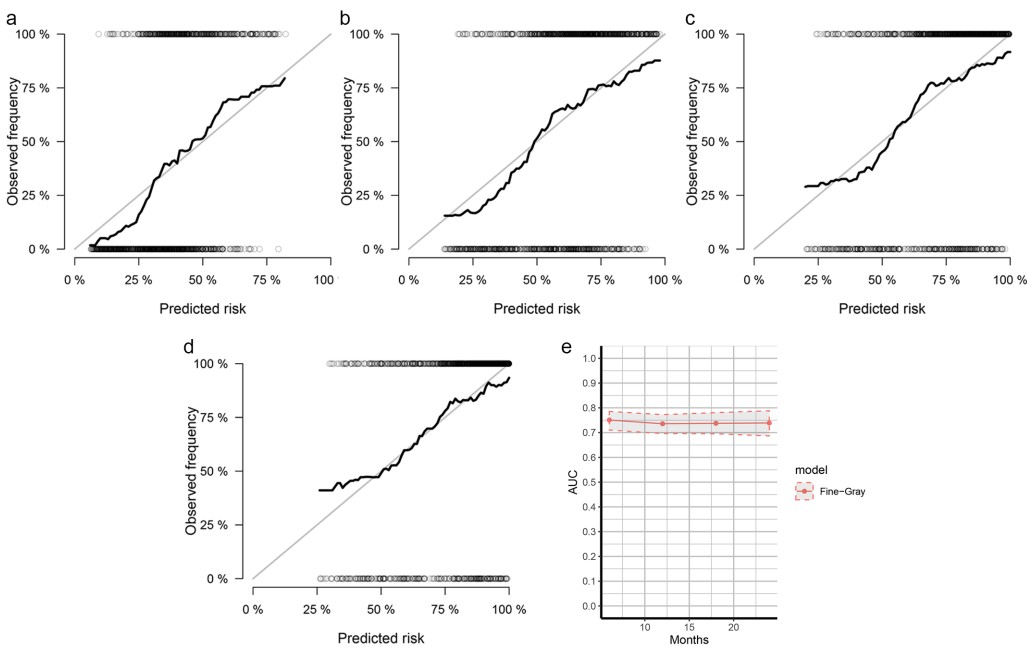

**Figure 3** **Calibration curves and AUC plots for internal validation of SH based nomograms.** The calibration plots for 6 months (A), 12 months (B) , 18 months (C) and 24 months (D) cancer specific death prediction of VEOPC patients; The $x$-axis shows the nomogram predicted probability while the $y$-axis is the actual survival estimated by the SH method, the black thick line overlaps the grey line indicating near perfect calibration; (E) time dependent AUC plots of SH-based nomograms.

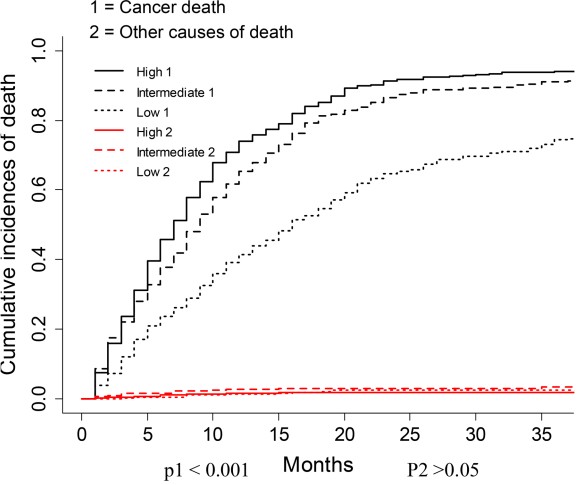

**Figure 4** **CIF plots based on the competing nomogram scores for the VEOPC patients.** The whole cohort of VEOPC patients were divided into three equally distributed groups. The groups were named "High", "Intermediate" and "Low" according to the levels of patients' competing nomogram scores.

### Funding

This research was supported by grants from the National Natural Science Foundation of China (81502386, 81772944, 91740106) and the High level health innovative talents program in Zhejiang and Natural Science Foundation of Zhejiang (LZ17H60003). The funders had no role in study design, data collection and analysis, decision to publish, or preparation of the manuscript.

### Grant Disclosures

The following grant information was disclosed by the authors:
National Natural Science Foundation of China: 81502386, 81772944, 91740106.
High level health innovative talents program in Zhejiang.
Natural Science Foundation of Zhejiang: LZ17H60003.

### Competing Interests

The authors declare there are no competing interests.

### Author Contributions

- Dongjun Dai conceived and designed the experiments, performed the experiments, analyzed the data, prepared figures and/or tables, authored or reviewed drafts of the paper, and approved the final draft.
- Yanmei Wang, Xinyang Hu and Hongchuan Jin analyzed the data, authored or reviewed drafts of the paper, and approved the final draft.
- Xian Wang conceived and designed the experiments, authored or reviewed drafts of the paper, and approved the final draft.

### Data Availability

Raw data was collected from the Surveillance, Epidemiology, and End Results (SEER) database and is available in Dataset S1.

### Supplemental Information

Supplemental information for this article can be found online at http://dx.doi.org/10.7717/peerj.8412#supplemental-information.

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
