# Peer review of "Prognostic analysis of very early onset pancreatic cancer: a population-based analysis"

_PeerJ, doi:10.7717/peerj.8412_

## Round 0.1 · original submission · Minor Revisions

This is an interesting and well-designed study, however the few descriptive and experimental issues raised by the reviewers should be addressed. English is not satisfactory, therefore re-editing should be seriously considered before publication.

Reviewer 1 ·

Basic reporting

1, the language of the article still need to be improved.
2, all p value are missed in these tables.
3, was Cumulative incidences of death conducted in R? what is the package you used in this process.

Experimental design

1,the sample from the TCGA database is too small.
2, there is no exploration of any molecular mechanism

Validity of the findings

the explanation of the results of your research could be better.

Additional comments

I reviewed the manuscript. The work is quite interesting and provides an epigenetic insight into PDAC.
In general, the data presented are of good quality and largely support the conclusions.

Reviewer 2 ·

Basic reporting

Clear, unambiguous, professional English (NOT satisfactory)
language used throughout.
Intro & background to show context. (YES)
Literature well referenced & relevant. (YES)
Structure conforms to PeerJ standards,
discipline norm or improved for clarity.
Figures are relevant, high quality, well
labelled & described. (NOT satisfactory)
Raw data supplied (see PeerJ policy). (YES)

Experimental design

Original primary research within Scope of (YES)
the journal.
Research question well defined, relevant (YES)
& meaningful. It is stated how the
research fills an identified knowledge gap.
Rigorous investigation performed to a (YES)
high technical & ethical standard.
Methods described with sufficient detail & (YES)
information to replicate.

Validity of the findings

Introduction and discussion should be elaborated, and more literature need to be cited.

Additional comments

Dai et al developed a nomogram for predicting the death caused by very early onset
19 pancreatic cancer (VEOPC)

1. Introduction is short and authors should elaborate and give details of all the factors that play a role in pathogenesis of pancreatic cancer.
2. Manuscript is badly written and English language is not good. There are many spelling mistakes. It seems authors are in hurry to send the manuscript. For example, line 46 “Duo”. The first sentence of introduction itself does not make any sense “Pancreatic cancer (PC) patients are estimated with a 5-year survival lower than 10%.
3. Line 144 r package can be changed to “R” package.
4. Line 161-165 is poorly written and very confusing. Author should consider it for rephrasing.
5. Line 180 the whole sentence is confusing
6. Author should provide list of abbreviations used in manuscript
7. figure 4 e should be in brackets (e)
8. Table 3 is not clear
9. Figure legend should be in detail

Reviewer 3 ·

Basic reporting

Overall: The authors have presented work similar to what they have done for other cancers (breast). Using SEER data presents some problems as other factors that might be associated with prognosis including tumor and germline genetic factors and behavioral factors (smoking, alcohol consumption which seem like they would be important in a competing risk model) are not available. The TCGA data were inadequate for identifying RNA based molecular differences. A similar paper regarding competing risk analysis and nomogram development has been published for patients with resected pancreatic cancer of the head with clear explanation of the value of competing risk analysis and the justification and utility of competing risk models and nomograms http://www.jcancer.org/v09p3156.htm. A more detailed discussion and justification for using competing risks especially in this young age group, is needed.

The varying use of age groupings was confusing. The analysis is described in the abstract as a comparison of very young to older patients (older is undefined while it is clear that younger is <45 years) and this is how the data are presented in Table 1 (patient characteristics). Then for the MV and prognostic analyses (Table 2) the older age group is then sub categorized into 4 groups that are compared with the very young. Finally the nomogram analyses are limited to the very young but in two age groups within this group (Table 3 and Fig 4). None of these groupings are justified or discussed making it difficult to understand who these patients are and what is the primary message being conveyed. It would be helpful if Table 1 provided patient characteristics for all age groupings or a supplemental table should be provided.

The grammar and sentence construction needs work. Someone who speaks English as a first language or is fluent in written English needs to help with the re-write. Awkward wording e.g. use of “involve” rather than “include” when talking about variables in the MV models, poor grammar and incomplete sentences e.g. “While the other studies found age was not a risk factor for PC patients.”, “And there was no significant prognostic difference between VEOPC and older PC in resectable PDAC.” are scattered throughout the manuscript. I note some but not all additional awkward grammar below in my critique.

Table headers need to be more descriptive and detailed. A table should be able to stand alone- without the paper. There is insufficient information in the table titles and table notes for a proper interpretation of the data being presented. Also, the figures need to be reformatted to address issues with hidden (overlapped with graphical presentation) and missing text ( labels).

Abstract:

Correct incomplete sentences:
“While the other studies found age was not a risk factor for PC patients.”, “And there was no significant prognostic difference between VEOPC and older PC in resectable PDAC.”

Define the older age group in age in years.


Introduction:
Line 43: The first sentence is awkwardly written.

Line 45: Should be “patients is 70 years”.

Line 46: “which was defined as very early onset PC (VEOPC)3.” Is this for this paper or is this standard terminology? In the next paragraph it is stated that definition of “young” has varied widely.

Line 49: wrong verb tense

Line 60: “To our best knowledge” is awkward phrasing.

Line 64: typo in “United States”

Line 65: “includes” instead of “involves”- this should be checked and corrected throughout.

Line 66: grammatically awkward and incorrect phrasing “Since pancreatic adenocarcinoma (PDAC) is the main type of PC, which has a worse prognosis than other histology types of PC and is frequently occurred in old patients, we choose the PDAC patients as our target cohort.”


Line 152: Reword subheading “Differentially analyses of miRNA and mRNA between VEOPC and older PC patients”


Line 163: The race data have not been presented this way in the text or the table. Be consistent.


Tables (as noted above need to be more detailed)

Table 1. Define acronyms, round % to 1 digit after the decimal if >1%, indent under main headings to make groupings easier to read/distinguish, add p-value for univariate comparison, add total N at tops of columns.

Correct the total number of unmarried

Table 2: What variables are included in the MV model? How was the "best" model selected? Age groups >70 are very similar and could be collapsed based on these results.

Table 3: Why this age grouping? How many patients are included in each group (as the data have not been presented previously)?

Under LNR- what is the maximum size i.e. 0.4 - ?

Figure 1: Reposition the legend as it overlaps with the curves. It is unclear what the red and black curves are- is that what is meant by the "right 1 and 2 represent..."?

Figure 4:

A more detailed description including the relevancy of these curves is needed for the reader to better understand what they are looking at. Competing risk analyses are not common and use of calibration curves for model prediction is probably not familiar to many readers.

Experimental design

Line 59: This is opinion and not backed up. Why is a nomogram needed? PDAC has few treatment options so it is difficult to see this being translated into clinical care (also this is a complex nomogram). Is it perceived that this would be useful for risk stratification in clinical trial recruitment, QOL interventions, future basic science or population based studies? Not clear.

Study variables and endpoints

Line 96: What are “small categorical variables”?

Line 97: Justify the groupings for marital status –especially as this is carried forward into the analyses within the VEOPC only, is statistically significant and included in the nomogram. Literature indicates that widowed younger people die earlier and in general are more likely to be single. This suggests that the married /not married grouping might not be the best for analyses within VEOPC.

Line 105- What was the stage distribution in the older PDAC patients?

Statistical Analysis:

Line 129- State whether the p-value is two-sided

Prognostic analysis of age in PDAC patients

Given that age is assessed in multiple groups (unexplained groupings) it seems table 1 would benefit by presenting data with this same grouping for "older" patients or at least provided as a supplemental table.

Line 147: It is not stated what other variables are included in the MV model.

miRNA and mRNA analyses

Given the small sample size and inability to conduct meaningful analyses of miRNA and mRNA this information is unhelpful and does not contribute anything meaningful to the analysis. The results mention that only 3 differentially expressed miRNAs and 17 differentially expressed mRNAs were found but it does not state the comparison groups. At least 3 different age group analyses were conducted so this is unclear. It could be mentioned in the discussion that this assessment was of interest but cannot be conducted at this time with data currently available.

Table 3 analyses
Given “Other” race is statistically significant and that most patients in this group are likely to be Asian (and thus driving the association) “Asians” should be broken out into their own category (for all analyses).

Validity of the findings

Abstract:
Reword this sentence “We observed VEOPC had better cancer specific survival (CSS) than older PC in patients with unresectable PDAC.” Does this mean both groups had unresectable PDAC or just the older patients?

Conclusion: Clarify that the first sentence pertains the VEOPC compared with older patients and that the second sentence pertains only to the VEOPC patients.

Overall
The relevance and importance of the nomogram is needed - other than it has not been done. A nomogram is a tool and it is not clear where the nomogram would be applied and why.

Statements and conclusions about the overall results of VEOPC vs all older PDAC, VEOPC vs older PDAC groups and within the VEOPC group need to be made extremely clear. Multiple analyses have been done which causes confusion when statements are not qualified according to the specific group of comparisons.

Weaknesses have been noted but it also should be noted that only factors available in the SEER registry were examined. The population to whom these results can be generalized needs to be made very clear.

The discussion and conclusion for the miRNA and mRNA analyses do not fit with the results section presentation- specifically there is too much teext devoted to this in the discussion. The results mention that only 3 differentially expressed miRNAs and 17 differentially expressed mRNAs were found (but not between what groups of patients). No other findings were statistically significant. A lot of discussion and speculation is given to these very underwhelming and limited findings- more so than to other aspects of the analysis. Additionally results are presented in the discussion that are not presented in the results section i.e. the specific miRNA and mRNAs that were identified. There is no segue that connects the discussion of PAF1 to the RNA analyses nor why PAF1 is relevant to the miRNA and mRNA results from this analysis.

---

## Round 0.2 · accepted · Accept

The authors have satisfactorily addressed the issues raised by the reviewers.

Reviewer 2 ·

Basic reporting

English is clear and understandable. Literature references are sufficient.

Experimental design

Experimental design are under the scope of Journal. Methods section is in detail.

Validity of the findings

Conclusions are well stated.

Additional comments

Thank you authors for revising the manuscript.